# Ca^2+^ Signaling in Cardiac Fibroblasts: An Emerging Signaling Pathway Driving Fibrotic Remodeling in Cardiac Disorders

**DOI:** 10.3390/biomedicines13030734

**Published:** 2025-03-17

**Authors:** Francesco Moccia, Antonio Totaro, Germano Guerra, Gianluca Testa

**Affiliations:** Department of Medicine and Health Sciences “V. Tiberio”, University of Molise, 86100 Campobasso, Italy; antonio.totaro@unimol.it (A.T.); germano.guerra@unimol.it (G.G.); gianluca.testa@unimol.it (G.T.)

**Keywords:** cardiac fibroblasts, myocardial fibrosis, Ca^2+^ signaling, InsP_3_ receptors, store-operated Ca^2+^ entry, STIM and Orai, receptor-operated Ca^2+^ entry, TRP channels, Piezo1 channels

## Abstract

Cardiac fibrosis is a scarring event that occurs in the myocardium in response to multiple cardiovascular disorders, such as acute myocardial infarction (AMI), ischemic cardiomyopathy, dilated cardiomyopathy, hypertensive heart disease, inflammatory heart disease, diabetic cardiomyopathy, and aortic stenosis. Fibrotic remodeling is mainly sustained by the differentiation of fibroblasts into myofibroblasts, which synthesize and secrete most of the extracellular matrix (ECM) proteins. An increase in the intracellular Ca^2+^ concentration ([Ca^2+^]_i_) in cardiac fibroblasts is emerging as a critical mediator of the fibrogenic signaling cascade. Herein, we review the mechanisms that may shape intracellular Ca^2+^ signals involved in fibroblast transdifferentiation into myofibroblasts. We focus our attention on the functional interplay between inositol-1,4,5-trisphosphate (InsP_3_) receptors (InsP_3_Rs) and store-operated Ca^2+^ entry (SOCE). In accordance with this, InsP_3_Rs and SOCE drive the Ca^2+^ response elicited by G_q_-protein coupled receptors (G_q_PCRs) that promote fibrotic remodeling. Then, we describe the additional mechanisms that sustain extracellular Ca^2+^ entry, including receptor-operated Ca^2+^ entry (ROCE), P2X receptors, Transient Receptor Potential (TRP) channels, and Piezo1 channels. In parallel, we discuss the pharmacological manipulation of the Ca^2+^ handling machinery as a promising approach to mitigate or reverse fibrotic remodeling in cardiac disorders.

## 1. Introduction

Cardiac fibrosis represents the most common outcome of multiple cardiovascular disorders (CVDs), including coronary artery disease (CAD), acute myocardial infarction (AMI), ischemic cardiomyopathy, dilated cardiomyopathy, diabetic cardiomyopathy, hypertensive heart disease, inflammatory heart disease, acute and chronic myocarditis, and aortic stenosis [1,2]. The myocardial structure basically consists of contractile cells (cardiomyocytes) and non-contractile cells surrounded by the cardiac extracellular matrix (ECM). The structural components of the cardiac ECM comprise fibrillar collagen, mainly type I fibers, heavier and thicker, and type III fibers, more flexible, and elastin, fibronectin, glycoproteins, proteoglycans, and glycosaminoglycans [3]. Such components contribute to the role of mechanical scaffold and contractile force transmission exerted by cardiac ECM, and the evenness between its deposition and degradation is crucial to the heart’s structural and functional integrity. This balance is tightly regulated by a system of matrix metalloproteinases (MMPs) whose alteration promotes myofibroblast transdifferentiation, mainly by transforming growth factor β1 (TGFβ1) signaling and ECM switch towards accumulation [3].

Cardiac fibrosis is driven by the excessive ECM accumulation that occurs in response to a severe myocardial injury and can be classified into “replacement fibrosis” and “reactive fibrosis” [1,2]. Replacement fibrosis is responsible for the formation of an organized (or compact) scar that replaces the cardiomyocytes that have been lost after a severe ischemic insult, e.g., AMI, and prevents the rupturing of the heart wall. Such fibrotic scar is composed of non-contractile, not-elastic tissue that does not contribute to force generation and pump activity, thereby resulting in heart failure (HF) with reduced ejection fraction (HFrEF) [4]. An alternate mode of fibrosis, known as “reactive fibrosis”, may occur in the absence of a large-scale cardiomyocyte loss or in non-ischemic cardiomyopathies [5]. Two distinct modes of “reactive fibrosis”, interstitial and perivascular, which often coexist, were described. Interstitial fibrosis is caused by the excessive deposition of collagen-rich ECM in the myocardial interstitium, is associated with the hypertrophic growth of cardiomyocytes, and is mainly driven by chronic stressors, such as post-AMI, hypertension, valve diseases, and pro-fibrotic systemic conditions. Perivascular fibrosis can be caused by the deposition of an excessive amount of connective tissue around coronary microvessels and is triggered by endothelial injury and coronary microvascular dysfunction [6]. Interstitial fibrosis is not primarily responsible for contractility reduction but may increase myocardial stiffness and reduce left ventricular compliance, thereby interfering with the diastolic function and leading to HF with preserved ejection fraction (HFpEF) [5]. On the other hand, perivascular fibrosis may precipitate diastolic function, thus worsening the outcome of an ischemic insult by narrowing the inner lumen of coronary vessels and favoring endothelial dysfunction [4,7].

Fibrosis also constitutes a pivotal substrate of arrhythmogenesis by interfering with the propagation of the rhythmic excitation signal [1,8]. The compact scar in the infarct tissue may function as an insulated, not-excitable area that anchors and converts re-entrant arrhythmia to sustained ventricular tachycardia. The interstitial fibrosis in the border zone and uninjured myocardium may separate the cardiomyocyte sheets by forming a network of non-conducting collagenous fibers, thereby resulting in reentrant tachycardia by inducing focal ectopic activity or by slowing or interrupting conduction. Furthermore, fibrosis-induced arrhythmias could also be driven by direct myofibroblast-cardiomyocyte coupling via gap junctions composed of connexin 43 [1,8]. Accordingly, cardiac fibrosis may predispose HF patients to sudden cardiac death independently of its effect on the ejection fraction [9].

Several pharmacological approaches have been developed to treat HFrEF patients, including β-blockers, sodium-glucose co-transporter 2 (SGLT2) inhibitors, and drugs targeting the renin-angiotensin-aldosterone system and Angiotensin Receptor Neprilysin Inhibitors (ARNI) [4]. Conversely, an effective strategy to mitigate or reverse fibrotic remodeling in cardiac diseases, including HFpEF, is still missing and requires understanding the molecular mechanisms driving ECM deposition. The primary cellular event that is responsible for the development of cardiac fibrosis is the differentiation of fibroblasts into myofibroblasts, which synthesize and secrete most of the ECM proteins (Figure 1). Myofibroblasts express contractile proteins, such as α-smooth muscle actin, and exhibit increased proliferative, migratory, and secretory features. Cardiac myofibroblasts produce large amounts of structural and matricellular ECM proteins and regulate matrix remodeling by secreting cytokines, growth factors, MMPs and tissue inhibitors of metalloproteinases (TIMPs) [4,7,10]. The fibrogenic signaling cascade can be triggered by multiple pro-inflammatory cytokines, such as tumor necrosis factor alpha (TNFα), TGFβ1, interleukin-1 (IL-1), and interleukin-6 (IL-6), hormones, such as noradrenaline, and vasoactive mediators, such as endothelin-1 (ET-1), angiotensin II (Ang II), A- and B-type natriuretic peptides (ANP and BNP) [4,7,10]. In addition, the transition of quiescent fibroblasts into active ECM-producing myofibroblasts can be stimulated by mechanical stretch, oxidative stress, and inflammatory signals released into the injured myocardium [4,7,10]. An increase in intracellular Ca^2+^ concentration ([Ca^2+^]_i_) lies at the heart of the intricate network of signaling pathways that regulate fibrinogenic signaling in both cardiac fibroblasts and myofibroblasts [11,12,13,14,15]. It has, therefore, been proposed that targeting the Ca^2+^ handling machinery could provide an alternative strategy to treat fibrosis-associated cardiac disorders, such as HFpEF [11,12,13,14,16,17]. Herein, we illustrate how intracellular Ca^2+^ signals are generated and contribute to fibroblast differentiation and cardiac fibrosis. We first focus our attention on the Ca^2+^ release pathways from intracellular stores, i.e., inositol-1,4,5-trisphosphate (InsP_3_) receptors (InsP_3_Rs) and two-pore channels (TPCs), which are, respectively, located in the ER and acidic vesicles. We then discussed the most important fibroblast Ca^2+^ entry routes, i.e., store-operated Ca^2+^ entry (SOCE), Transient Receptor Potential (TRP) and Piezo1 channels, but we also described the ionotropic P2X receptors. In parallel, we discuss a number of drugs either already approved by the Food and Drug Administration (FDA) system or under pre-clinical investigation, which could be exploited to manipulate the Ca^2+^ handling machinery to mitigate or reverse fibrotic remodeling in cardiac disorders. The most suitable molecular targets include SOCE, some TRP isoforms, and Piezo1 channels.

## 2. Ca^2+^ Signaling in Cardiac Fibroblasts Induced by Physiological Agonists

An increase in [Ca^2+^]_i_ with defined spatial and temporal patterns regulates a multitude of cellular functions, including proliferation, migration, differentiation, metabolism, gene expression, and ROS stress adaptation [18,19,20,21]. Intracellular Ca^2+^ signals can be generated by the plethora of chemical cues that are present in the pro-fibrotic environment of the myocardial interstitium and activate G_q_-protein coupled receptors (G_q_PCRs). A crucial role in fibroblast Ca^2+^ signaling is played by the polymodal TRP channels, which can integrate pro-inflammatory signals, oxidative stress and enhanced mechanical load, thereby promoting myofibroblast differentiation. Furthermore, emerging evidence also suggests that the mechanosensitive Piezo1 channels are required to sense the progressive ECM stiffening that occurs during cardiac repair [11,12,13,16]. In the present section, we mainly discuss the role of InsP_3_Rs and SOCE in fibroblast Ca^2+^ signaling that occurs downstream of G_q_PCRs. In Section 3, we illustrate the function of TRP channels, while in Section 4, we focus on Piezo1 channels.

### 2.1. An Introduction to Cardiac Fibroblast Ca^2+^ Signaling

Early work showed that the resting [Ca^2+^]_i_ in human cardiac fibroblasts (HCFs) ranges between 30 and 80 nM and is maintained by a sophisticated network of Ca^2+^-transporting proteins, including Sarco-Endoplasmic Reticulum Ca^2+^ ATPase (SERCA), Na^+^/Ca^2+^ exchanger (NCX), plasma membrane Ca^2+^-ATPase (PMCA), and mitochondria (Figure 2) [22]. A transcriptomic analysis revealed that the following isoforms are expressed in HCFs: SERCA1, SERCA2, SERCA3, NCX3, PMCA1, PMCA3, and PMCA4 [22]. Mitochondria buffer cytosolic Ca^2+^ through the Ca^2+^ uniporter complex in the inner mitochondria membrane, which includes the mitochondrial Ca^2+^ uniporter (MCU) and several accessory subunits, such as mitochondrial calcium uptake 1–3 (MICU1–3), that interact with MCU to form a functional channel that rapidly responds to a local increase in cytosolic Ca^2+^ levels [23]. Mitochondrial Ca^2+^ uptake via MICU1-dependent MCU gating could be involved in fibroblast-to-myofibroblast differentiation [24,25]. A minority (about 30%) of HCFs show spontaneous Ca^2+^ oscillations that could be resumed by challenging the cells with fetal bovine serum (FBS) [22].

FBS can trigger asynchronous oscillations in [Ca^2+^]_i_ by promoting Ca^2+^ release from the endoplasmic reticulum (ER) through InsP_3_Rs and SOCE across the plasma membrane (Figure 2) [26,27]. For those who are not familiar with the Ca^2+^ signaling machinery, any agonist leading to phospholipase C (PLC) activation and InsP_3_ production does not only cause ER Ca^2+^ release through InsP_3_Rs but also activates SOCE (see Section 2.2) [18,19,28]. Several PLC isoforms have been described [18], including PLCβ and PLCγ, which are activated, respectively, by G_q_PCRs and Receptor Tyrosine Kinases (RTKs). FBS leads to InsP_3_ production by activating a mixture of RTKs on the plasma membrane [26,27]. In accordance with this notion, FBS-induced Ca^2+^ oscillations in HCFs are shaped by rhythmic ER Ca^2+^ mobilization through InsP_3_Rs and sustained by SOCE [22]. Conversely, fibroblast growth factor-23 (FGF-23) stimulates a rapid increase in [Ca^2+^]_i_ that then rapidly decays to a long-lasting plateau phase by activating FGF receptor 1 [29]. Intracellular Ca^2+^ oscillations, as well as biphasic Ca^2+^ signals, can also be elicited upon the activation of multiple G_q_PCRs [11]. Compelling evidence demonstrates that the increase in [Ca^2+^]_i_ that arises downstream of either RTKs and G_q_PCRs, including FGF receptor 1 [29], vascular endothelial growth factor receptor 2 (VEGR-2) [30], Ang II receptor type 1 (AT1) [31,32], P2Y2 receptors [33,34,35], bradykinin receptor type 1 (BR1) [36], and Ca^2+^-sensing receptor (CaSR) [37], recruits pro-fibrotic signaling pathways and could therefore be critical for cardiac fibrosis [11]. A representative list of physiological agonists and membrane receptors that elicit intracellular Ca^2+^ signals in cardiac fibroblasts has been provided in Table 1.

### 2.2. Intracellular Ca^2+^ Release in Cardiac Fibroblasts

The ER is the largest intracellular Ca^2+^ reservoir in mammalian cells, although the nuclear envelope, endo-lysosomal (EL) vesicles, and Golgi apparatus can also function as an endogenous Ca^2+^ reservoir [18,19,42]. HCFs express all three known InsP_3_R isoforms, i.e., InsP_3_R1, InsP_3_R2 and InsP_3_R3, while ryanodine receptors (RyRs), which are predominant in cardiomyocytes [43], are absent [22]. Several agonists have been shown to induce InsP_3_ production, including Ang II, bradykinin, ATP and UTP [34,40]. The three InsP_3_R subtypes exhibit a different sensitivity to InsP_3_ and cytosolic Ca^2+^, which can prime InsP_3_Rs to InsP_3_-dependent activation. Therefore, the spatiotemporal profile of intracellular Ca^2+^ signals is highly influenced by the underlying InsP_3_R(s) [20,44]; InsP_3_R2 is the most sensitive isoform to InsP_3_ and Ca^2+^, thereby sustaining long-lasting Ca^2+^ oscillations; InsP_3_R1 presents an intermediate sensitivity to InsP_3_ and supports either transient Ca^2+^ signals or irregular Ca^2+^ spikes; InsP_3_R3, which shows the less sensitivity to InsP_3_ and Ca^2+^, serves as an anti-oscillatory unit that only produces monophasic increases in [Ca^2+^]_i_. Therefore, InsP_3_R2 is the most likely source of rhythmic Ca^2+^ release during the spontaneous Ca^2+^ oscillations that were detected in HCFs. This hypothesis is supported by the evidence that the spiking Ca^2+^ activity was abolished by the pharmacological blockade of phospholipase C (PLC) and InsP_3_Rs and by the depletion of the ER Ca^2+^ pool [22]. Future work is required to assess why some agonists trigger intracellular Ca^2+^ oscillations while others induce biphasic Ca^2+^ signals in cardiac fibroblasts. Distinct InsP_3_R isoforms could be engaged by different agonists or even by the same agonist in cardiac fibroblasts deriving from diverse cardiac chambers or species (Table 1), as recently shown in mouse vs. human cerebrovascular endothelial cells [45,46,47,48]. Alternately, distinct G_q_PCR membrane receptors could undergo a differential regulation by protein kinase C (PKC), as demonstrated in rat hepatocytes [49].

The EL Ca^2+^ pool is emerging as a novel regulator of intracellular Ca^2+^ dynamics in the cardiovascular system (Figure 2) [42]. Lysosomal Ca^2+^ can be released through TPCs and TRP Mucolipin 1–3 (TRPML1–3). TPCs can trigger InsP_3_-induced Ca^2+^ release in vascular endothelial cells [45,50,51] and circulating endothelial colony-forming cells [52,53] and type 2 RyR (RyR2) activation in cardiomyocytes [54,55] and vascular smooth muscle cells (VSMCs) [56,57,58]. TRPML1 also mobilizes the ER Ca^2+^ pool in several cell types [59,60], including vascular endothelial cells [61] and VSMCs [62]. The transcripts encoding for TRPML1–3 were recently detected in HCFs [63], but their functional role is unclear. Similarly, it is still unknown whether TPCs, which present two isoforms in mammals [42], are expressed in cardiac fibroblasts. TPC1 and TPC2 proteins were recently detected in human cardiac mesenchymal stromal cells (C-MSCs) [64], from which HCFs are derived [65]. TPCs are gated by the second messenger, nicotinic acid adenine dinucleotide phosphate (NAADP), which triggers ER Ca^2+^ release through InsP_3_Rs followed by SOCE activation and promotes FBS-induced intracellular Ca^2+^ oscillations in C-MSCs [64]. Future work will have to confirm whether TPCs are also expressed in cardiac fibroblasts and participate in pro-fibrotic signaling.

### 2.3. Extracellular Ca^2+^ Entry in Cardiac Fibroblasts

The Ca^2+^ response to chemical cues is sustained over time by Ca^2+^ entry from the extracellular milieu. As described in Section 3.3, agonist binding to G_q_PCRs leads to InsP_3_-induced ER Ca^2+^ release, which results in a drop in the ER Ca^2+^ concentration ([Ca^2+^]_ER_), which signals the activation of a Ca^2+^-entry pathway, known as SOCE, on the plasma membrane [11,66]. In addition, agonist-induced Ca^2+^ signals can be maintained by receptor-operated Ca^2+^ entry (ROCE), which is supported by members of the TRP sub-family (see Section 3) that are gated upon the activation of the PLC pathway [13,67,68], and ionotropic P2X receptors [69]. Finally, cardiac fibroblasts express a number of mechanosensitive channels that translate subtle changes in the ECM stiffness into a Ca^2+^ signal, which is primarily mediated by Piezo1 channels (see Section 4) [70,71].

#### 2.3.1. SOCE

SOCE is a ubiquitous Ca^2+^ entry pathway that is activated upon the InsP_3_-dependent depletion of the ER Ca^2+^ pool to reload the ER with incoming Ca^2+^ and thereby maintain biphasic and oscillatory Ca^2+^ signals over a prolonged period [28,66,72,73]. SOCE is activated when a reduction in the [Ca^2+^]_ER_ activates STIM (STIM1 and STIM2), a single-pass transmembrane protein that senses changes in intraluminal Ca^2+^ levels and directly gates three distinct Ca^2+^ entry pathways on the plasma membrane: Orai (Orai1, Orai2, and Orai3), which mediates the Ca^2+^ release-Ca^2+^ activated current (I_CRAC_); TRP Canonical 1 (TRPC1) and/or TRPC4, which mediate the non-selective store-operated current (I_SOC_); and a heteromeric complex consisting of Orai1, TRPC1, and TRPC4 and displaying a Ca^2+^ over Na^+^ selectivity intermediate between the I_CRAC_ and I_SOC_, known as I_CRAC_-like [28,66,72,73,74,75,76]. Transcriptomic analysis showed that STIM1 and Orai1-3 are expressed in HCFs (Figure 2) [22]. Immunoblotting revealed that STIM1 and Orai1 are also present at protein levels in HCFs [77] and mouse cardiac fibroblasts [31]. Subsequent studies detected the expression of STIM1 and Orai1 proteins in human ventricular [39,78] and atrial [29] fibroblasts. The transcripts encoding for TRPC1 and TRPC4 and the corresponding proteins were also found in cardiac fibroblasts from several species [11,67]. Ca^2+^ imaging recordings confirmed that SOCE can be activated by the pharmacological (with thapsigargin) and physiological (with Ang II, thrombin or FGF-23) depletion of the ER Ca^2+^ store in HCFs [22,63], human ventricular fibroblasts [39,78], human atrial fibroblasts [29], and mouse cardiac fibroblasts [31,79].

However, the biophysical characterization of the store-operated currents in cardiac fibroblasts is still missing: it is still unknown whether they express the I_CRAC_, I_SOC_ or I_CRAC_-like current [22,63]. SOCE is sensitive to 2-aminoethoxydiphenyl borate (2-APB) in HCFs [32] and atrial cardiac fibroblasts [29], whereas it is inhibited by SKF96365 in rat cardiac fibroblasts [31]. These drugs have long been regarded as Orai1 inhibitors, but they are not specific, and no conclusion about the Ca^2+^-permeable channel mediating SOCE can be drawn based solely on their use [28,80]. Thus far, the critical role of the I_CRAC_ in SOCE has only been shown in rat cardiac fibroblasts, in which the genetic deletion of STIM1 or Orai1 significantly reduced Ang II-induced Ca^2+^ entry [31]. In agreement with this finding, Ang II- and thrombin-induced SOCE was not downregulated in cardiac fibroblasts isolated from TRPC1- and TRPC4-deficient mice [79] but was inhibited by the selective Orai1 inhibitor, GSK7975A [81]. Therefore, the prevailing view that SOCE is primarily mediated by STIM1 and Orai1 in cardiac fibroblasts is supported by these (and others, see below) preliminary pieces of evidence but needs to be clearly demonstrated in human ventricular and atrial fibroblasts [11,67,73].

SOCE was up-regulated in ventricular cardiac fibroblasts deriving from HF patients due to an increase in Orai1 expression, thereby resulting in cardiac fibrosis through an increase in collagen secretion [78]. The pharmacological blockade of SOCE with YM58483/BTP-2, which is a rather selective Orai1 antagonist [80,82,83], reduced collagen secretion, thereby suggesting the therapeutic inhibition of SOCE could provide an alternative strategy to treat myocardial fibrosis in HF [78]. Consistent with this observation, Ang II- and thapsigargin-induced SOCE were significantly increased in aged human ventricular fibroblasts, while there was no difference in the ER Ca^2+^ content as well as in STIM1 and Orai1 expression [39]. Therefore, SOCE could also stimulate cardiac fibrosis during aging. Furthermore, the pharmacological blockade of SOCE with 2-APB and lithium, which is emerging as a novel Orai1 inhibitor [84,85], reduced Ang II-induced collagen synthesis in HCFs [32]. Similarly, SKF96365 and the genetic silencing of Orai1 reduced Ang II-induced overexpression of fibronectin, connective tissue growth factor, and α-SMA in rat cardiac fibroblasts [31]. 

Moreover, SOCE was found to stimulate fibrosis by inducing the nuclear translocation of the Ca^2+^-sensitive transcription factor, nuclear factor of activated T cells (NFAT), isoform NFATc4, which promotes the expression of pro-inflammatory cytokines, and by boosting TGFβ1 expression [31]. Finally, the chemotherapeutic drug, doxorubicin, which may induce HF as an off-target effect in cancer patients [86], induced apoptosis in HCFs by activating Orai1-mediated Ca^2+^ entry, which in turn led to ROS production, early apoptosis, and cell cycle arrest in the G2 phase [77]. The profibrotic effect of doxorubicin, both in vitro and in vivo, was blocked by YM58483/BTP-2 and the genetic silencing of Orai1 [77]. Therefore, SOCE is emerging as a crucial regulator of cardiac fibrosis, and Orai1 might provide a promising therapeutic target for preventing or mitigating cardiac fibrosis in HF. Intriguingly, many Food and Drug Administration (FDA)-approved small-molecule drugs have been repurposed as Orai1 inhibitors, including flecainide [14,65], propranolol [87], lithium [84,85], lansoprazole, tolvaptan, and roflumilast [88]. As anticipated above, confirming the involvement of STIM1 and Orai1, as well as investigating the role of STIM2, Orai2 and Orai3 [89,90], in human fibrosis is mandatory to effectively translate SOCE into therapeutic applications.

#### 2.3.2. ROCE

ROCE is mediated by Ca^2+^-permeable channels that primarily open in response to G_q_PCR activation and sustain Ca^2+^ entry through store-operated channels [13,67,68]. In principle, SOCE is physiologically gated by agonist-induced depletion of the ER Ca^2+^ store but is regarded as a distinct Ca^2+^ entry mode as it can be activated independently on agonist binding to membrane receptors (e.g., by pharmacologically depleting the ER Ca^2+^ store with thapsigargin or cyclopiazonic acid) [28,66]. ROCE is mediated by some members of the TRP superfamily of non-selective cation channels and, therefore, will be described in Section 3.

#### 2.3.3. Ionotropic P2X Receptors

P2X receptors are ligand-gated non-selective cation channels that open in response to ATP binding, thereby mediating membrane depolarization and extracellular Ca^2+^ entry (Figure 2) [91,92,93]. Among the seven P2X isoforms described in mammalian cells (P2X1–7) [91,92,93], P2X4 and P2X7 receptors were detected in HCF at both mRNA [22] and protein [33] levels. An early investigation showed that ATP-induced proliferation and migration were impaired by genetic silencing of either P2X4 or P2X7. These findings were mimicked by abolishing P2Y2 signaling, which involves an InsP_3_-induced ER Ca^2+^ release [33]. A more recent study confirmed that P2X7 is critical in mediating the abnormal activation of cardiac fibroblasts and cardiac fibrosis in a mouse model of transverse aortic constriction (TAC) and in TGFβ1-hyperstimulated rat cardiac fibroblasts [94]. These findings were supported by the evidence that TAC activates sympathetic efferent nerves that release ATP, which in turn stimulates P2X7 receptors on cardiomyocytes, coronary endothelial cells and cardiac fibroblasts [95]. P2X7 receptors, in turn, drive the assembly of NLRP3 (nucleotide-binding domain, leucine-rich-containing family, pyrin domain-containing 3) inflammasome and IL-1β production, thereby favoring cardiac fibrosis [95]. Nevertheless, it is a decrease in intracellular K^+^ concentration rather than the increase in [Ca^2+^]_i_ to promote NLRP3 activation [96]. As pointed out in [11], further work is required to characterize the role of P2X4 and P2X7 receptors in ATP-induced membrane currents and Ca^2+^ signals in cardiac fibroblasts and to validate them as promising targets to prevent or mitigate cardiac fibrosis.

## 3. TRP Channels

The mammalian TRP superfamily of non-selective cation channels consists of 28 members that are divided into six sub-families according to their sequence homology: canonical (TRPC1–7), melastatin (TRPM1–8), vanilloid (TRPV1–6), ankyrin (TRPA1), polycystin (TRPP), and TRPML1–3. The TRPP sub-family includes eight members, but only TRPP2, TRPP3, and TRPP5 function as true ion channels [97,98,99]. TRP channels mainly reside on the plasma membrane, but they may also be found in intracellular organelles, such as ER (TRPC1, TRPV1, and TRPM8) [100,101], mitochondria (TRPV1) [102,103], and lysosomes (TRPML1–TRPML3, TRPM2, and TRPA1) [42,104,105]. The majority of TRP channels are permeable to both Na^+^ and Ca^2+^, thereby promoting membrane depolarization and Ca^2+^ entry [97,98,99], which may be followed or not by Ca^2+^-induced Ca^2+^ release from the ER through InsP_3_Rs [106,107,108].

TRP channels serve as polymodal cellular sensors that can integrate a myriad of chemical and physical cues deriving from both the surrounding microenvironment and the metabolic activity, including second messengers [such as diacylglycerol (DAG) and arachidonic acid (AA)], reactive oxygen species [such as H_2_O_2_ and 4-Hydroxynonenal (4-HNE)], intracellular ions (such as an elevation in cytosolic Ca^2+^ and a reduction in cytosolic Mg^2+^), dietary agonists (such as menthol, capsaicin, and allyl isothiocyanate (AITC) or AITC), gasotransmitters (such as nitric oxide and hydrogen sulfide), G-proteins, deformations of the plasma membrane (such as osmotic swelling and membrane stretch), and changes in temperature [97,98,109,110,111,112,113,114,115,116]. TRP channels are widely expressed in cardiac fibroblasts (Table 2, Table 3 and Table 4) and mediate an influx of Ca^2+^ that can recruit multiple pro-fibrotic signaling pathways [11,67,70].

### 3.1. TRPC Channels and ROCE

TRPC channels are primarily responsible for ROCE in both excitable and non-excitable cells [109,117] and all seven TRPC isoforms (TRPC1–7) were detected in cardiac fibroblasts (Table 2). As anticipated in Section 3.2.1, TRPC1 and TRPC4 are activated by STIM1 in response to the depletion of the ER Ca^2+^ store, but they do not seem to mediate SOCE in cardiac fibroblasts. TRPC3, TRPC6, and TRPC7 are gated by the intracellular second messenger, DAG [109,118,119], whereas TRPC4 and TRPC5 are sensitive to both DAG and Gα_i/o_ proteins [120,121].

**Table 2 biomedicines-13-00734-t002:** TRPC channels in cardiac fibroblasts.

TRPC Isoform	Cell Type and Related Reference	Associated Cardiac Disorder and Related Reference
TRPC1	HCFs [22,122]; human atrial fibroblasts [123]; mouse cardiac fibroblasts [79,123]; rat cardiac fibroblasts [124,125]	Unknown
TRPC3	Mouse cardiac fibroblasts [79]; mouse atrial fibroblasts [126]; rat cardiac fibroblasts, human, goat, and canine atrial fibroblasts [127]; human ventricular cardiac fibroblasts [128]; rat atrial fibroblasts [129]; rat ventricular fibroblasts (PMID: 17204501)	Atrial fibrillation [126,127] Pressure overload-induced heart failure [130]
TRPC4	HCFs [22,63,122]; mouse cardiac fibroblasts [79,123]	Unknown
TRPC5	Mouse cardiac fibroblasts [79]; rat ventricular fibroblasts [124]	Unknown
TRPC6	HCFs [22,63,122], human right ventricular fibroblasts [131], mouse cardiac fibroblasts [79,132], rat cardiac fibroblasts [125,133], rat cardiac fibroblasts [134]	Cardiac wound healing after injury [132] Pressure overload-induced heart failure [131,135]
TRPC7	Mouse cardiac fibroblasts [79]; rat neonatal cardiac fibroblasts [136]	Unknown

Abbreviation: HCFs—human cardiac fibroblasts.

#### 3.1.1. TRPC3

TRPC3 channels are involved in cardiac fibrosis and fibrosis-associated cardiac disorders, including atrial fibrillation [126,127,129,137] and pressure overload-induced HF [128,130]. TRPC3 protein was up-regulated in atria deriving from AF patients as well as from goat and canine models of AF. TRPC3-mediated Ca^2+^ entry was enhanced in atrial fibroblasts from AF dogs, thereby promoting proliferation and survival by engaging the extracellular signal-related kinase-1/2 (ERK-1/2) pathway. Notably, the pharmacological blockade of TRPC3 with Pyr3 reduced proliferation and ECM deposition in left atrial fibroblasts from AF dogs. Furthermore, the in vivo administration of Pyr3 mitigated the development of the AF substrate in the canine model by reducing fibroblast proliferation and fibrotic differentiation [127]. Consistent with this finding, an increase in the plasma levels of homocysteine, which can be regarded as an independent risk factor for atrial fibrillation [138], has been associated with TRPC3-dependent cardiac fibrosis. A recent investigation suggested that homocysteine binds to G_q_PCRs, thereby activating TRPC3, which in turn physically interacts with and inhibits the antifibrotic protein sirtuin-1 [126]. The pharmacological blockade of TRPC3-mediated Ca^2+^ entry with another pyrazole derivative, i.e., Pyr-10, reduced homocysteine-induced TGFβ1 secretion from mouse atrial fibroblasts [126]. This observation supports the view that TRPC3 provides a promising target to rescue myocardial fibrosis and treat AF.

TRPC3 protein was also up-regulated in human ventricular cardiac fibroblasts isolated from HF patients (both HFpEF and HFrEF), thereby causing a robust increase in Ang II-evoked Ca^2+^ entry and NFATc3-dependent myofibroblast differentiation [128]. Furthermore, the genetic knockdown and the pharmacological blockade of TRPC3 with Pyr10 or polyphenols reduced cardiac fibrosis in a mouse model of pressure overload-induced HF [128]. The in vivo administration of Pyr3 also attenuated pressure overload-induced maladaptive fibrosis in mouse hearts [130]. Excess left ventricular diastolic filling also activates TRPC3 in mouse ventricular cardiomyocytes, thereby leading to the Ca^2+^-dependent recruitment of NADPH oxidase 2 (Nox2) and aberrant ROS production. The functional interplay between TRPC3 and Nox2 was found to be critical for HF development [139]. Additionally, local Ca^2+^ entry through TRPC3 could be induced by TGFβ1 through a yet-to-be-elucidated mechanism and lead to Nox2-dependent ROS production in mouse cardiac fibroblasts [130]. A follow-up investigation revealed that mechanical stress is translated into myocardial fibrosis by the recruitment of the GEF-H1-RhoA signaling pathway. In accord, the Nox2-dependent ROS burst promoted microtubule depolymerization, which activated the microtubule-associated Rho guanine nucleotide exchange factor, GEF-H1. GEF-H1, in turn, engaged RhoA to mediate actin cytoskeletal reorganization-dependent transcription of profibrotic genes [130]. Therefore, blocking TRPC3 could mitigate myocardial fibrosis associated with several cardiac disorders. 

Pyr3 is the most selective TRPC3 inhibitor [140,141], but it can exert toxic effects in vivo, is hydrolyzed into the inactive acid derivative Pyr8, and can also inhibit Orai1 [142,143]. Therefore, a battery of novel TRPC3 antagonists was designed to interfere with TRPC3-mediated Ca^2+^ entry in both neurological and cardiovascular pathologies [143,144,145]. These small-molecule drugs include compound 20, which derives from Pyr3, JW-65 and compound 60a, which display a strong selectivity towards TRPC3 with respect to other TRP isoforms. Preclinical and clinical studies will have to assess whether these novel drugs can be exploited to treat myocardial fibrosis.

#### 3.1.2. TRPC6

TRPC6 channels play a crucial role in fibroblast proliferation and differentiation [11,67]. An early report showed that endothelin-1-induced rat myofibroblast differentiation was inhibited by TRPC6-mediated constitutive Ca^2+^ entry, which induced the nuclear translocation of NFAT [125]. Conversely, TGFβ1 and Ang II stimulated proliferation by inducing a robust influx of Ca^2+^ in HCFs that was likely to be mediated by TRPC6 and sustained by the reverse (Ca^2+^ entry) mode of NCX [122]. Likewise, TRPC6-mediated Ca^2+^ entry recruited calcineurin to promote TGFβ1- and Ang II-dependent mouse myofibroblast differentiation both in vitro and in vivo [132]. In accordance with this, TGFβ1 and Ang II induced an increase in TRPC6 protein expression through the p38 MAPK and serum response factor (SRF) signaling pathways [132]. Then, TRPC6-mediated Ca^2+^ influx promoted myofibroblast differentiation and stimulated dermal and cardiac wound healing by recruiting calcineurin [132]. A subsequent investigation demonstrated that endoglin, which serves as a co-receptor for TGFβ1 [146], is required to maintain TGFβ1-dependent TRPC6 activation and TRPC6-mediated calcineurin recruitment in a mouse model of right ventricular pressure overload [131]. Additionally, TRPC6 was involved in collagen gene expression upon the engagement of the profibrotic transcription factor, Yes-associated protein (YAP), in Ang II-stimulated rat cardiac fibroblasts [133]. TRPC6 further supported cardiac fibrosis associated with sympatho-βAR (β-adrenoceptor) activation. In this setting, the proinflammatory and profibrotic mediator, galectin-3, induced the over-expression of the Ca^2+^-activated K^+^ channel, K_Ca_3.1, thereby resulting in a significant hyperpolarization of mouse cardiac fibroblasts [147]. The ensuing increase in the driving force for Ca^2+^ entry enhanced TRPC6- as well as TRPV4-dependent Ca^2+^ signals, thereby favoring myofibroblast differentiation. Interestingly, Ca^2+^ entry through TRP channels boosted K_Ca_3.1 activation [147], as also reported in vascular endothelial cells [20].

These findings indicate that TRPC6 may also serve as a promising target to mitigate myocardial fibrosis. Consistently, the orally bioactive TRPC6 inhibitor, BI 749327, was found to reduce the expression of profibrotic genes and blunt myocardial fibrosis in mice exposed to sustained pressure overload [135]. Similarly, blocking the expression of TRPC6 with the bioactive compound Tinglu Yixin granule rescued cardiac function and tempered myocardial fibrosis in diabetic mice [148]. Additional small-molecule drugs that could be exploited to inhibit TRPC6 but are yet to be probed in cardiac fibroblasts include congeners of the labdane diterpene (+)-larixol [149], a component from *Larix decidua* turpentine, and the more stable larixyl-6-carbamate [150] and N-methylcarbamate SH045 [151] as well as several synthetic compounds, such as BCTC, TC-I 2014, and SAR 7334 [152].

### 3.2. TRPV Channels

The TRPV subfamily includes six isoforms, known as TRPV1-TRPV6, although the Ca^2+^-selective TRPV5 and TRPV6 are primarily responsible for Ca^2+^ absorption in the kidney and intestine [13,67]. TRPV1-TRPV4 are non-selective cation channels that present a permeability ratio P_Ca_/P_Na_ ranging between ~1 and ~15 and, therefore, may induce both membrane depolarization and robust Ca^2+^ signals [99]. TRPV1–4 channels are truly polymodal channels that can integrate thermal, mechanical and chemical stimuli, including temperature changes, membrane stretch, arachidonic acid, epoxyeicosatrienoic acids, ROS and a variety of dietary compounds [99,153,154]. Although they are commonly thought to be expressed on the plasma membrane, TRPV1, TRPV2, and TRPV3 channels may also reside in endogenous organelles, such as ER and mitochondria [155]. Emerging evidence suggests that TRPV1, TRPV3, and TRPV4 channels play a crucial role in myocardial fibrosis (Table 3).

**Table 3 biomedicines-13-00734-t003:** TRPV channels in cardiac fibroblasts.

TRPV Isoform	Cell Type and Related Reference	Associated Cardiac Disorder and Related Reference
TRPV1	Mouse cardiac fibroblasts [156,157,158,159]	Reduces pressure overload-induced cardiac fibrosis [157] Reduces high-salt-induced cardiac fibrosis [158] Reduces isoproterenol-induced cardiac fibrosis [159] TAC-induced cardiac fibrosis [160]
TRPV2	HCFs [63]; rat cardiac fibroblasts [161]	Unknown
TRPV3	HCFs [63]; rat cardiac fibroblasts [162]	Pressure overload-induced cardiac fibrosis [162]
TRPV4	HCFs [63]; human ventricular fibroblasts [163]; rat cardiac fibroblasts [164,165]; mouse cardiac fibroblasts [166]	Maladaptive fibrotic remodeling after AMI [166] Diabetes-induced cardiac fibrosis [165] TAC- and isoproterenol-induced cardiac fibrosis [167,168]

Abbreviations: HCFs—human cardiac fibroblasts; TAC—transverse aortic constriction.

#### 3.2.1. TRPV1

The role of TRPV1 in myocardial fibrosis has been a matter of controversy [11]. A preliminary report showed that the genetic deletion of TRPV1 enhanced post-AMI fibrosis and reduced myocardial contractility by stimulating the TGFβ1/Smad2 signaling pathway [156]. Conversely, a subsequent investigation demonstrated that capsaicin, a selective dietary agonist of TRPV1 [110,169], ameliorated reduced cardiac fibrosis in pressure overload mice by activating TRPV1-dependent Ca^2+^ signals. In accord, capsaicin failed to exert its cardioprotective effect on TRPV1 knocked-out (KO) mice [157]. Similarly, dietary capsaicin consumption selectively mitigated long-term high-salt diet-induced cardiac hypertrophy and fibrosis in TRPV1-expressing mice [158]. Moreover, capsaicin attenuated post-AMI inflammation and myocardial fibrosis by activating TRPV1 in diabetic mice [170]. In accordance with these findings, isoproterenol-induced myocardial fibrosis was significantly attenuated in transgenic mouse models overexpressing TRPV1 [159]. Furthermore, it has been shown that TRPV1-mediated Ca^2+^ signals recruit cardioprotective signaling pathways in several mouse models of AMI and HF [171,172,173], whereas the genetic deletion of TRPV1 exacerbated cardiac hypertrophy, myocardial inflammation and fibrosis in a mouse model of TAC [160]. However, other investigations argued against the cardioprotective role of TRPV1. The subcutaneous administration of the TRPV1 inhibitor, BCTC (4-(3-Chloro-2-pyridinyl)-N-[4-(1,1-dimethylethyl)phenyl]-1-piperazinecarboxamide), reduced cardiac hypertrophy and HF in vivo [168]. Additionally, TRPV1 was up-regulated and enhanced with severe cardiac fibrosis in hypertrophic hearts isolated from transgenic mice overexpressing the catalytic subunit α of protein phosphatase 2A α [174]. These discrepancies in the cardioprotective outcome of TRPV1 activation could reflect the widespread expression of TRPV1 [11,175], which could be expressed in cardiac fibroblasts, cardiomyocytes, coronary endothelial cells, and afferent adrenergic fibers [70]. For instance, TRPV1-expressing sympathetic adrenergic fibers have long been known to stimulate arrhythmogenic ventricular remodeling in response to an acute ischemic event [176] or TAC [177]. Therefore, future work is mandatory to understand whether the pharmacological manipulation of TRPV1 represents a suitable strategy to interfere with myocardial fibrosis.

#### 3.2.2. TRPV3

TRPV3 expression has been detected in cardiac fibroblasts from several species (Table 3). In rat cardiac fibroblasts, TRPV3 stimulation by the natural monoterpoid carvacrol promoted cell proliferation and increased TGFβ1 expression [162]. Additionally, carvacrol induced an increase in [Ca^2+^]_i_ that was dampened by ruthenium red [162], although ruthenium red is not a selective TRPV3 antagonist [178]. Similarly, genetic silencing of TRPV3 significantly reduced Ang II-induced intracellular Ca^2+^ signals [162]. The mechanism by which Ang II activates TRPV3 has not been elucidated, but it is worth noting that G_q_PCRs may gate TRPV3 via PLCβ-mediated hydrolysis of PIP_2_ [179]. The in vivo administration of carvacrol exaggerated fibrotic remodeling in mouse models of pressure-overloaded rats, whereas ruthenium red mitigated carvacrol-induced cardiac dysfunction [162]. Therefore, these preliminary findings suggest that TRPV3 might play a role in myocardial fibrosis, although they need to be confirmed using TRPV3 KO mice [11]. Furthermore, more selective TRPV3 antagonists, such as PC5 and Trpvcin [180], should be probed to assess whether targeting TRPV3 protects the heart from fibrotic remodeling.

#### 3.2.3. TRPV4

TRPV4 is critical to cardiac fibroblast differentiation into myofibroblasts by integrating mechanical cues and TGFβ1-derived signals (Table 3) [11,70,181]. An early report demonstrated that 4α-phorbol 12,13-didecanoate (4αPDD), a rather selective TRPV4 agonist [182], induced a non-selective cation current and intracellular Ca^2+^ signals in rat cardiac fibroblasts, which were inhibited by the genetic silencing of TRPV4 protein [161]. A subsequent investigation revealed that TGFβ1-induced myofibroblast differentiation was sensitive to the pharmacological (with AB159908) and genetic (with a small interfering RNA selectively targeting TRPV4 expression) blockade of TRPV4 [164]. Consistently, TGFβ1 treatment increased TRPV4 expression and TRPV4-mediated Ca^2+^ signals. Furthermore, TGFβ1-induced myofibroblast differentiation was enhanced by increasing the ECM stiffness [164]. This evidence is consistent with the known ability to sense the ECM environmental rigidity by physically interacting with β1 integrins [183,184,185].

Intriguingly, TRPV4-mediated Ca^2+^ signals were also up-regulated by TGFβ1 in human ventricular fibroblasts and promoted myofibroblast differentiation by recruiting the MAPK/ERK signaling pathway [163]. A follow-up investigation revealed that the genetic deletion of TRPV4 protected the heart from adverse fibrotic remodeling after an AMI and confirmed that cardiac fibroblasts deficient in TRPV4 failed to differentiate on high-stiffness ECM substrates even in the presence of saturating concentrations of TGFβ1 [166]. Consistently, TRPV4 up-regulation in cardiac fibroblasts supported diabetic myocardial fibrosis by activating the TGF-β1/Smad3 signaling pathway in rats [165]. The crucial role played by TRPV4 in pathological remodeling has been recently confirmed by Yáñez-Bibe and coworkers, who showed that the genetic deletion of TRPV4 prevented adverse fibrotic remodeling in two different mouse models of HF (i.e., TAC- and isoproterenol-induced HF) [167]. These authors further demonstrated that TRPV4-mediated Ca^2+^ entry was required to up-regulate TRPC6 expression through the Ca^2+^-dependent recruitment of calcineurin [167]. Therefore, the pharmacological blockade of TRPV4 represents a promising strategy to therapeutically interfere with myocardial fibrosis and fibrosis-associated cardiac disorders [186]. Notably, the selective TRPV4 antagonist, GSK2798745, has succeeded in entering clinical trials (Phase I and Phase II) as a drug candidate for the treatment of several disorders (https://clinicaltrials.gov accessed on 22 January 2025), including HF (NCT02119260). GSK2798745 is regarded as the most selective TRPV4 antagonist [187]. Other TRPV4 blockers, which are more selective than GSK2798745 and therefore did not enter clinical trials, include GSK2193874, HC-067047, RN-1734, and berberine, the plant alkaloid [53,188,189].

### 3.3. TRPM Channels

The TRPM sub-family includes eight members, known as TRPM1-TRPM8, which were named based upon the first identified member, i.e., TRPM1, initially regarded as a tumor suppressor in melanoma [190]. TRPM channels were classified into four subgroups according to their sequence homology: (1) TRPM1/TRPM3, (2) TRPM4/TRPM5, (3) TRPM6 and TRPM7, and (4) TRPM2 and TRPM8. With the exception of TRPM4 and TRPM5, which are only permeable to monovalent cations, TRPM channels serve as non-selective cation channels that regulate both the membrane potential and the [Ca^2+^]_i_ [190]. TRPM2, TRPM4 and TRPM7 emerged as the sole TRPM isoforms involved in myocardial fibrosis (Table 4) [11,191].

**Table 4 biomedicines-13-00734-t004:** TRPM channels in cardiac fibroblasts.

TRPV Isoform	Cell Type and Related Reference	Associated Cardiac Disorder and Related Reference
TRPM2	Rat cardiac fibroblasts [192]	Unknown
TRPM4	HCFs [63]; human left ventricular fibroblasts [193]; human and mouse atrial fibroblasts [194]	HF [193]
TRPM7	HCFs [63]; human atrial fibroblasts [123]; rat ventricular cardiac fibroblasts [195,196]	Sino-atrial fibrosis [197]

Abbreviations: HCFs—human cardiac fibroblasts; HF—heart failure.

#### 3.3.1. TRPM2

TRPM2 is a non-selective cation channel that serves as a redox biosensor and, depending on the extent of the oxidative stress, may either engage an anti-oxidant program or lead to Ca^2+^-dependent cell death [198]. An early report found that TRPM2 was expressed and mediated non-selective cation currents in rat cardiac fibroblasts. Additionally, TRPM2 expression and TRPM2-mediated Ca^2+^ signaling were increased by exposing cardiac fibroblasts to hypoxia [192]. This finding suggests that TRPM2 may be involved in adverse fibrotic remodeling after an ischemic event, but future work is mandatory to confirm this hypothesis.

#### 3.3.2. TRPM4

TRPM4 is a non-selective monovalent cation channel that can be activated by an increase in [Ca^2+^]_i_ [190,199]. As expected, single-channel recordings first demonstrated that a sub-membrane Ca^2+^ pulse was able to activate TRPM4 in both human and mouse atrial fibroblasts [194]. The pharmacological (with the rather selective inhibitor, 9-phenanthrol) and genetic (by exploiting TRPM4 KO mice) of TRPM4 activity strongly reduced cardiac fibroblast cell growth and myofibroblast differentiation [194]. Moreover, TRPM4 expression and TRPM4 currents were significantly up-regulated in cardiac fibroblasts isolated from the left ventricle of HF patients [193]. Consistently, pre-treatment with TGFβ1 increased the protein levels of TRPM4 and TRPM4 currents in healthy cardiac fibroblasts [193], thereby confirming that TRPM4 may play a crucial role in myocardial fibrosis.

#### 3.3.3. TRPM7

TRPM7 is a constitutively open non-selective cation channel that can be regulated by multiple feedback mechanisms, including the intracellular Mg^2+^ concentration and PIP_2_, and bears a significant permeability to divalent cations, including Ca^2+^, Mg^2+^, and Zn^2+^ [191]. TRPM7-mediated currents were initially recorded in rat ventricular fibroblasts and found to be inhibited by PLCβ-dependent hydrolysis of PIP_2_ [195]. Endogenous TRPM7 currents were also measured in mouse and human atrial fibroblasts [123,200]. An early report showed that atrial fibroblasts from HF patients presented a strong increase in TRPM7 currents and TRPM7-mediated Ca^2+^ signals [123]. Exposure to TGFβ1 induced TRPM7 up-regulation in healthy atrial fibroblasts, and TRPM7 was, in turn, required to sustain TGFβ1-induced myofibroblast differentiation [123]. Similar findings were reported in mouse cardiac fibroblasts [123]. Likewise, TRPM7 expression was dramatically enhanced in Ang II-treated rat cardiac fibroblasts, and the genetic deletion of TRPM7 with either a selective short hairpin RNA or small interfering RNA interfered with Ang II-induced myofibroblast differentiation [196,201,202]. The same result was obtained by blocking TRPM7 with 2-APB [196], but 2-APB is not a selective TRPM7 antagonist [80]. Consistent with these findings, sino-atrial node fibrosis in rats was associated with the Ang II/TRPM7/Smad2 signaling pathway, thereby providing the first in vivo evidence that TRPM7 may be involved in myocardial fibrosis [197]. TRPM7 is not a canonical biosensor of oxidative stress [21,198]. However, TRPM7 was found to support H_2_O_2_-induced Ca^2+^ entry and myofibroblast differentiation by recruiting the ERK1/2 signaling pathways in rat cardiac fibroblasts [203]. It should, however, be noted that a recent investigation revealed that H_2_O_2_ inhibits TRPM7 by targeting the reactive thiols, Cys-1809 and Cys-1813, which are located in the Zn^2+^-binding motif [204]. These findings strongly suggest that TRPM7 is involved in myocardial fibrosis. Nevertheless, extending these observations to animal models of myocardial fibrosis associated with HF or diabetes is strongly recommended to place TRPM7 among the novel molecular targets to treat fibrosis-associated cardiac disorders. Interestingly, the synthetic sphingosine analog FTY720, which has been approved by the FDA for the treatment of multiple sclerosis, has been shown to inhibit endogenous TRPM7 currents in cardiac fibroblasts [205].

### 3.4. TRPA1

TRPA1 is the sole member of the TRPA sub-family and serves as a polymodal channel being sensitive to changes in temperature, mechanical deformation, and multiple chemical cues, including ROS, hydrogen sulfide, and electrochemical chemical activators that are abundant in pungent compounds [116,206,207]. TRPA1 is expressed in cardiac fibroblasts from multiple species (Table 5), and the role of TRPA1-mediated Ca^2+^ signals in myocardial fibrosis is still controversial [16,207,208]. TRPA1 activation ameliorated right ventricular fibrosis by stimulating CGRP release in a mouse model of PAH. CGRP can then generate autocrine signals that interfere with TGFβ1-induced myofibroblast differentiation via NF-κB activation [209]. Likewise, TRPA1 deficiency accelerated cardiac dysfunction and myocardial fibrosis in a transgenic mouse model of diabetes-induced dilated cardiomyopathy [210] and in aged mice [211]. Dietary compounds, such as AITC and cinnamaldehyde, proved to be effective at stimulating TRPA1 both in vitro and in vivo and, therefore, could be exploited for therapeutic purposes [209,210,212]. However, other investigations provided contrasting results showing that TRPA1 deficiency protects from TGFβ1-induced myofibroblast differentiation and may provide a valuable target for fibrosis-associated cardiac disorders [213,214,215,216]. For instance, Wang and coworkers showed that blocking TRPA1 with the selective inhibitors, HC-030031 and TCS-5861528, inhibited pressure overload-induced myocardial fibrosis by preventing the Ca^2+^-dependent recruitment of calcineurin and CaMKII [214].

Furthermore, TRPA1 promoted TGFβ1-induced myofibroblast differentiation in a mouse model of AMI. TRPA1-mediated Ca^2+^ signals recruited calcineurin to promote the nuclear translocation of NFAT, which in turn engaged the transdifferentiation-related gene expression program [213]. The protective effect of TRPA1 deficiency on myocardial interstitial fibrosis was also reported in diabetic rats [215]. The discrepant role of TRPA1 could reflect the different involvement of cardiac fibroblasts in cardiac repair after AMI, hypertension, and aging [2,217]. Additionally, the stage of myocardial fibrosis and the activation of immune cells should also be taken into consideration [16,216,218]. It has been shown that TRPA1 stimulation causes macrophages to differentiate towards the M2 phenotype, thereby accelerating the infiltration of M2 macrophages in the heart, which exacerbates pressure-overload-induced myocardial fibrosis [214].

**Table 5 biomedicines-13-00734-t005:** TRPA1 expression in cardiac fibroblasts.

TRPA1	Cell Type and Related Reference	Associated Cardiac Disorder and Related Reference
TRPA1	HCFs [212,219]; human cardiac fibroblasts [215]; rat cardiac fibroblasts [209,215]; mouse cardiac fibroblasts [213]	Endogenous suppressor of cardiac fibrosis in PAH [209] Stimulates pressure overload-induced cardiac fibrosis [214] Maladaptive fibrotic remodeling after AMI [213] Diabetes-induced cardiac fibrosis [215]

Abbreviations: HCFs—human cardiac fibroblasts; PAH—pulmonary arterial hypertension.

## 4. Piezo Channels

Piezo1 and Piezo2 are non-selective channels that are mainly located on the plasma membrane and serve as primary mechanotransduction devices in mammalian cells [220], including cardiac fibroblasts (Table 6) [70]. While Piezo2 is far more abundant in sensory neurons, Piezo1 plays a crucial role in cardiovascular mechanosensation [221]. Piezo1 channels mediate membrane depolarization and intracellular Ca^2+^ signals in response to a plethora of mechanical stimuli, including lateral membrane tension as well as membrane stiffness and roughness [70]. Therefore, Piezo1 channels are ideally suited to detect and finely adjust ECM remodeling during cardiac repair [70]. An early report showed that Piezo1 was expressed in mouse and human cardiac fibroblasts, thereby inducing an increase in [Ca^2+^]_i_ triggered by the selective Piezo1 agonist, Yoda1, and inhibited by the genetic deletion of Piezo1 protein [222]. Piezo1-mediated Ca^2+^ signals induced the secretion of the pro-fibrotic cytokine and IL-6 via the p38 MAPK signaling pathway. 

Furthermore, Piezo1 was responsible for the basal secretion of IL-6 from cardiac fibroblasts cultured on softer collagen-coated substrates [222]. Piezo1 was also sensitive to mechanical stimulation in human atrial fibroblasts [223]: a negative pressure pulse activated inward currents that were attenuated by the genetic deletion of Piezo1 with a selective siRNA and by GsMTx4, an established blocker of stretch-activated channels. Moreover, Piezo1 expression and activity were up-regulated in cardiac fibroblasts isolated from patients suffering from atrial fibrillation [223], thereby highlighting Piezo1 as a crucial player of myocardial fibrosis in arrhythmias. Consistently, Piezo1 was also found to mediate stretch-induced brain natriuretic peptide and TGFβ1 secretion from rat cardiac fibroblasts [224]. Interestingly, the mechanical activation of Piezo1 led to the activation of the large-conductance, Ca^2+^-activated K^+^ channels (BK_Ca_), which may further enhance Piezo1-mediated Ca^2+^ entry by hyperpolarizing the resting membrane potential [223]. A follow-up investigation demonstrated that Piezo1 interacts with β1 integrins to reorganize the F-actin cytoskeleton in a human atrial fibroblast cell line; furthermore, Piezo1 expression was required to enable atrial fibroblast adaptation to ECM stiffness [225].

Intriguingly, Piezo1 also induced an increase in the stiffness of adjacent fibroblasts by promoting IL-6 secretion [225]. The positive mechanobiological feedback loop between Piezo1 and β1 integrins was confirmed by an in silico study supporting the functional interaction between these signaling proteins during cardiac fibrotic remodeling [226]. These findings lent support to the view that Piezo1 channels could be targeted to treat fibrotic remodeling [70]. Consistent with this hypothesis, a transgenic mouse model overexpressing Piezo1 displayed cardiac hypertrophy and myocardial fibrosis that were associated with a significant increase in Piezo1-mediated Ca^2+^ signals in cardiac fibroblasts [227]. In addition to being localized on the plasma membrane, Piezo1 may also be detected in mitochondria, as recently reported in rat cardiac fibroblasts [71]. Mitochondrial Ca^2+^ release through Piezo1 channels may be induced by culturing cardiac fibroblasts on stiff substrates, although the physiological implications of this signal are unclear [71]. Future work will have to assess whether the mitochondrial Piezo1 regulates mitochondrial respiration in cardiac fibroblasts, as recently reported in vascular endothelial cells [228].

**Table 6 biomedicines-13-00734-t006:** Piezo1 expression in cardiac fibroblasts.

Piezo1	Cell type and related reference	Function or disease
Piezo1	HCFs [222]; human atrial fibroblasts [223]; rat cardiac fibroblasts [224]; mouse cardiac fibroblasts [222,227]	Myofibroblast differentiation [222] Adaptation to ECM stiffness [225] Cardiac fibrosis [227]

Abbreviations: HCFs—human cardiac fibroblasts; ECM—extracellular matrix.

## 5. Conclusions

Fibrotic remodeling is a hallmark of most cardiac disorders and may severely affect cardiac performance, ultimately leading to patients’ death. Understanding the intracellular signaling pathways driving cardiac fibroblast transition from quiescence to the activated state is required to design effective therapeutic approaches to mitigate or even reverse fibrotic remodeling in cardiac diseases, including HFpEF. An increase in [Ca^2+^]_i_ is emerging as a crucial mediator of fibroblast recruitment and activation as the Ca^2+^ signaling machinery is capable of integrating the chemical and physical cues generated by the pro-fibrotic environment of the myocardium interstitium. The available evidence suggests that several components of the Ca^2+^ toolkit endowed with cardiac fibroblasts provide promising and druggable targets to treat fibrosis-associated cardiac disorders. SOCE, TRPC3, TRPC6, TRPV4, TRPM7 and Piezo1 channels are emerging as candidates for the development of novel therapeutic approaches. SOCE and TRP channels are crucial for fibrotic remodeling. In vivo experiments showed that the pharmacological blockade of TRPC3 and TRPC6 mitigate cardiac fibrosis, thereby suggesting that their role is not redundant with that played by SOCE.

As anticipated in Section 2.3.1, it is mandatory to assess whether blocking SOCE also exerts an antifibrotic effect in vivo. It is worth noting that SOCE and TRPC3/TRPC7 are both activated downstream of PLC activation: this evidence supports the notion that they are coupled with distinct Ca^2+^-dependent effectors involved in the different steps of fibrotic remodeling. On the other hand, TRPV4 could interact with Piezo1 to sense any change in the ECM matrix: TRPV4 could be more sensitive to softened ECM [185], while Piezo1 could be more sensitive to stiffer substrates [229]. We believe that in vitro investigations will be instrumental to understanding whether distinct Ca^2+^ entry/Ca^2+^ release routes engage different effectors involved in cardiac fibrosis and whether the fibroblast Ca^2+^ signaling pathways leading to HFpEF or HFrEF are similar or differ somehow. In parallel, as mentioned throughout the text, the pro-fibrotic role of each component of the fibroblast Ca^2+^ toolkit has not been probed in animal models. It will be worth studying whether the fibrotic response is triggered by different conditions, i.e., AMI and hypertension, and whether or not it impinges on the same ion channels.

The pharmacological manipulation of SOCE will require the identification of its molecular makeup, but as explained in the text, many FDA-approved drugs are already available on the market. It will also be important to gain further insights into the pathogenic role of the different Ca^2+^ entry pathways by using in vivo models of the different cardiac pathologies, e.g., AMI, diabetic cardiomyopathy or hypertensive heart disease, involving fibrotic remodeling. We believe that future work will highlight the involvement of TPCs and P2X receptors in myocardial fibrosis as their pathogenic role is emerging in other diseases, including cancer, arrhythmias, neurodegenerative diseases and viral infections [42,92,230,231,232,233,234]. As discussed throughout the text, more selective small-molecule inhibitors are required to exploit the therapeutic potential of Ca^2+^ entry channels, such as the distinct TRP isoforms and Piezo1 channels. As mentioned above, the evidence that dysregulated Ca^2+^ signals are involved in cardiovascular, neurological, oncological, immune, and skeleton-muscle disorders is fostering the search for specific inhibitors of each component of the Ca^2+^ signaling toolkit. Intriguingly, it has recently been shown that empagliflozin, a SGLT2 inhibitor that reduces cardiac fibrosis currently in a Phase 3 clinical trial for the treatment of HFpEF [235,236], reduces cardiac fibrogenesis by inhibiting InsP_3_-induced ER Ca^2+^ release and SOCE in atrial fibroblasts [237]. This finding, although preliminary, further supports the view that the pharmacological manipulation of fibroblast Ca^2+^ signals may represent a promising strategy to treat cardiac fibrosis.

## Figures and Tables

**Figure 1 biomedicines-13-00734-f001:**
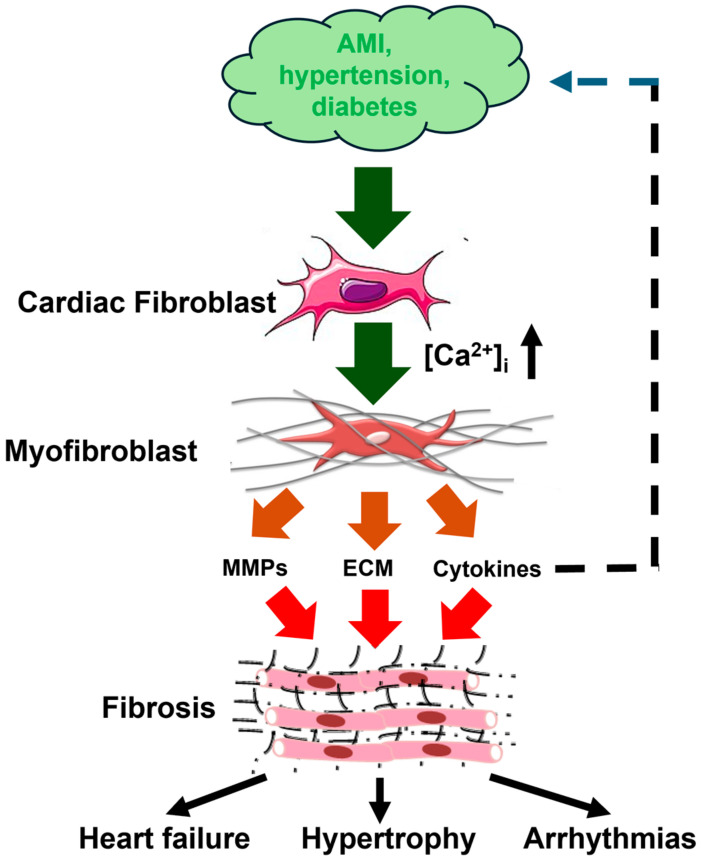
Cardiac fibrogenesis cascade and fibrosis-associated cardiac disorders. Pathological stimuli, such as AMI (acute myocardial infarction), hypertension, and diabetes, stimulate cardiac fibroblasts to differentiate into myofibroblasts through an increase in [Ca^2+^]_i_. Myofibroblasts are instrumental for fibrotic remodeling by synthesizing and secreting ECM (extracellular matrix) proteins, MMPs (matrix metalloproteinases), and cytokines, including IL-6 (interleukin-6) and transforming growth factor β1 (TGFβ1). Cardiac fibrotic remodeling may, in turn, lead to heart failure, cardiac hypertrophy, and arrhythmias. This figure is inspired by Figure 1 in [11].

**Figure 2 biomedicines-13-00734-f002:**
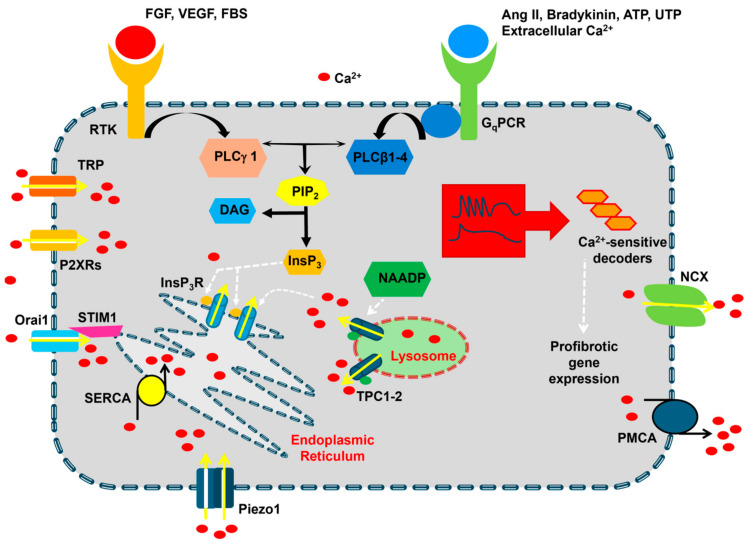
The Ca^2+^ signaling machinery in cardiac fibroblasts. The resting [Ca^2+^]_i_ in cardiac fibroblasts is maintained by the interplay between Ca^2+^-clearing mechanisms that are primarily located in the endoplasmic reticulum, i.e., Sarco-Endoplasmic Reticulum Ca^2+^-ATPase (SERCA), and on the plasma membrane, i.e., Na^+^/Ca^2+^ exchanger (NCX) and plasma membrane Ca^2+^-ATPase (PMCA). Extracellular agonists bind either G_q_-protein coupled receptors (G_q_PCRs) or Receptor Tyrosine Kinases (RTKs) to activate, respectively, phospholipase Cβ (PLCβ) and PLCγ, which cleave phosphatidylinositol 4,5-bisphosphate (PIP_2_) into diacylglycerol (DAG) and inositol-1,4,5-trisphosphate (InsP_3_). InsP_3_, in turn, causes Ca^2+^ release from the endoplasmic reticulum by activating InsP_3_ receptors (InsP_3_Rs). The following depletion of the endoplasmic reticulum Ca^2+^ store results in the activation of SOCE, which could be mediated by STIM1 and Orai1 proteins. InsP_3_-induced Ca^2+^ release could be triggered by nicotinic acid adenine dinucleotide phosphate (NAADP)-induced Ca^2+^ release from lysosomal vesicles, although this mechanism remains to be elucidated in cardiac fibroblasts. Ca^2+^ entry in cardiac fibroblasts can also occur via Transient Receptor Potential (TRP) channels, P2X receptors, and Piezo1 channels. The Ca^2+^ response to extracellular stimulation may occur as intracellular Ca^2+^ oscillations or biphasic Ca^2+^ signals, which may lead to the activation of a profibrotic gene expression program.

**Table 1 biomedicines-13-00734-t001:** Representative list of physiological agonists that induce intracellular Ca^2+^ signals in cardiac fibroblasts.

Agonist	Receptor	Cell Type(s)	Kinetics of the Ca^2+^ Signal	Reference(s)
FGF	FGF receptor 1	Human atrial fibroblasts	Biphasic	[29]
VEGF	VEGFR-2	Human atrial fibroblasts	Monotonic	[30]
Ang II	AT1	Rat cardiac fibroblasts	Biphasic	[31,38]
Ang II	AT1	Human ventricular fibroblasts	Biphasic	[39]
Ang II	AT1	HCFs	Biphasic	[32]
ATP	P2Y2	Rat cardiac fibroblasts	Transient	[40]
ATP	P2Y2	Rat ventricular fibroblasts	Transient	[35]
UTP	P2Y11	Rat ventricular fibroblasts	Biphasic	[41]
Bradykinin	B1R	Rat cardiac fibroblasts	Transient	[36,40]
Extracellular Ca^2+^	CaSR	Rat cardiac fibroblasts	Biphasic	[37]

Abbreviations: Ang II—angiotensin II; AT1—Ang II receptor type 1; B1R—bradykinin receptor type 1; CaSR—Ca^2+^-sensing receptor; VEGF—vascular endothelial growth factor.

## Data Availability

No new data were generated to prepare the present manuscript.

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
