# Peer review of "Ca2+ Signaling in Cardiac Fibroblasts: An Emerging Signaling Pathway Driving Fibrotic Remodeling in Cardiac Disorders"

_biomedicines, 2025, doi:10.3390/biomedicines13030734_

Round 1
Reviewer 1 Report
Comments and Suggestions for Authors
The manuscript addresses the role of calcium (Ca²⁺) signaling in cardiac fibroblasts, specifically its involvement in cardiac fibrosis, a key pathological feature in many cardiovascular diseases (CVDs). Cardiac fibrosis is a major contributor to heart failure and arrhythmogenesis, making this review quite relevant. The detailed focus on the mechanisms of Ca²⁺ signaling, including SOCE (store-operated Ca²⁺ entry), TRP (transient receptor potential) channels, and Piezo1, provides a good foundation for understanding the molecular basis of fibrosis and the potential for therapeutic targeting. The manuscript highlights emerging druggable targets.
The introduction provides sufficient background on the role of fibroblasts and myofibroblasts in extracellular matrix (ECM) deposition and the detrimental effects of fibrosis on cardiac function. The authors have adequately introduced the role of intracellular Ca²⁺ signaling in fibroblast activation and ECM remodeling. However, the introduction could benefit from a more concise summary of the specific objectives of the review, as it currently overlaps with the abstract in scope.
The authors have thoroughly analyzed and synthesized data from a wide range of studies, providing a detailed and critical assessment of the field. The literature cited spans both foundational and recent studies, reflecting an in-depth understanding of the topic. The use of tables to summarize findings (e.g., TRP channel involvement in cardiac fibrosis) is particularly effective in organizing and presenting complex data.
The conclusions are well-aligned with the review's objectives, emphasizing the importance of targeting Ca²⁺ signaling pathways in cardiac fibroblasts to mitigate fibrosis. The authors identify key knowledge gaps and propose directions for future research, such as the need for in vivo studies and the development of more selective pharmacological tools.
Minor Points:
- Clarify the specific objectives of the review to avoid redundancy with the abstract.
- Reduce overlapping content between the introduction and other sections.
- Ensure all figures and tables are fully labeled and easy to interpret. For instance, Figure 2 could include a legend explaining abbreviations like "InsP3R" and "SOCE" for clarity.
- Standardize abbreviations throughout the manuscript (e.g., "HFpEF" vs. "HFrEF") to maintain consistency.
- While the discussion of SOCE and TRP channels is thorough, the manuscript could provide additional mechanistic insights into how these pathways interact during fibrosis.
- Expand on the therapeutic implications of targeting specific channels (e.g., TRPC3, TRPV4).
- The authors briefly mention knowledge gaps but could elaborate on specific experimental approaches to address these.
Author Response
Dear Reviewer #1,
We truly thank you for your insightful comments on our manuscript entitled: “Ca2+ signaling in cardiac fibroblasts: an emerging signaling pathway driving fibrotic remodeling in cardiac disorders” for publication as Review Article in Biomedicines – Special Issue “The Role of Ion Channels and Transporters in Human Health and Diseases (2nd edition)”.
We carefully addressed all your concerns, which significantly improved the quality of our manuscript.
More specifically:
1) Clarify the specific objectives of the review to avoid redundancy with the abstract.
We do thank the Reviewer for these observations. We reworded the specific objectives as suggested by the Reviewer.
2) Reduce overlapping content between the introduction and other sections.
We do thank the Reviewer for this suggestion. We partially reworded the Introduction and the beginning of section 1 to reduce overlapping content between the introduction and other sections.
3) Ensure all figures and tables are fully labeled and easy to interpret. For instance, Figure 2 could include a legend explaining abbreviations like "InsP3R" and "SOCE" for clarity.
We do thank the Reviewer for this observation. We explained all the abbreviations in both Figure 1 and Figure 2.
4) Standardize abbreviations throughout the manuscript (e.g., "HFpEF" vs. "HFrEF") to maintain consistency.
We do thank the Reviewer for this recommendation. We carefully went through the manuscript again. All the abbreviations are standardized. If we miss something (which may happen when you read the same manuscript many, many times, as the Reviewer knows), we’ll fixed this issue during the proof corrections (if the manuscript is accepted for publication).
5) While the discussion of SOCE and TRP channels is thorough, the manuscript could provide additional mechanistic insights into how these pathways interact during fibrosis.
We do thank the Reviewer for this suggestion. We addressed this issue in the Conclusions (lines 706-715).
6) Expand on the therapeutic implications of targeting specific channels (e.g., TRPC3, TRPV4).
We do thank the Reviewer for this suggestion. We expanded upon the pharmacological blockade of TRPC3 (lines 417-420), TRPC6 (lines 452-457) and TRPV4 (lines 550-553) as suggested.
7) The authors briefly mention knowledge gaps but could elaborate on specific experimental approaches to address these.
We do thank the Reviewer for this suggestion. We addressed this issue in the Conclusions, lines 715-721.
We therefore hope that the manuscript will now be regarded worth of being published on this thrilling special issue of Biomedicines.
Sincerely,
Reviewer 2 Report
Comments and Suggestions for Authors
The authors present a very comprehensive review about Ca2+ signalling in cardiac fibroblasts, that complements and extends previous reviews.
Here are some suggestions that might improve this work:
1.) I would welcome it if the authors mention, that fibrosis is not only scaring, but has several physiological functions, like orchestration of the extracellular matrix . Furthermore, fibrosis may decrease the wall tension of the heart chambers if necessary. Therefore, any hampering of fibrosis may have detrimental effects.
2.) Abstract: “We focus our attention on the functional interplay between inositol-1,4,5-trisphosphate (InsP3) receptors (InsP3Rs) and store-operated Ca2+ entry (SOCE), which drives the Ca2+ response elicited by Gq-protein coupled receptors (GqPCRs) that lead to intracellular Ca2+ signals shaped by interaction between inositol-1,4,5-trisphosphate (InsP3) receptors (InsP3Rs) and store-operated Ca2+ entry (SOCE).
Please simplify this sentence.
3.) L26: “Then, we describe the additional mechanisms sustain extracellular Ca2+ entry, …”
Better: Then, we describe the additional mechanisms that sustain extracellular Ca2+ entry, …
4.) Figure 1: It is almoust identical to a figure to the figure of Feng et al. J. Cardiovasc. Dev. Dis. 2019. The authors must mention this.
5.) L 315: “The following influx of Ca2+ drives NLRP3 (nucleotide-binding domain, leucine-rich-containing family, pyrin domain-containing 3) inflammasome and IL-1b production, thereby favoring cardiac fibrosis [94].”
In the mentioned reference, Ca2+i was not measured. Furthermore, it is known that it is rather the K+ efflux than Ca2+ influx that activates NLRP3.
Author Response
Dear Reviewer #2,
We truly thank you for your insightful comments on our manuscript entitled: “Ca2+ signaling in cardiac fibroblasts: an emerging signaling pathway driving fibrotic remodeling in cardiac disorders” for publication as Review Article in Biomedicines – Special Issue “The Role of Ion Channels and Transporters in Human Health and Diseases (2nd edition)”.
We carefully addressed all your concerns, which significantly improved the quality of our manuscript.
More specifically:
1) I would welcome it if the authors mention, that fibrosis is not only scaring, but has several physiological functions, like orchestration of the extracellular matrix. Furthermore, fibrosis may decrease the wall tension of the heart chambers if necessary. Therefore, any hampering of fibrosis may have detrimental effects.
We do thank the Reviewer for these observations. We have discussed this issue in section 1, lines 38-48.
2) Abstract: “We focus our attention on the functional interplay between inositol-1,4,5-trisphosphate (InsP3) receptors (InsP3Rs) and store-operated Ca2+ entry (SOCE), which drives the Ca2+ response elicited by Gq-protein coupled receptors (GqPCRs) that lead to intracellular Ca2+ signals shaped by interaction between inositol-1,4,5-trisphosphate (InsP3) receptors (InsP3Rs) and store-operated Ca2+ entry (SOCE). Please simplify this sentence.
We do thank the Reviewer for this suggestion. We simplified the sentence as kindly requested.
3) L26: “Then, we describe the additional mechanisms sustain extracellular Ca2+ entry, …”. Better: Then, we describe the additional mechanisms that sustain extracellular Ca2+ entry, …”.
We do thank the Reviewer for this observation. We have reworded the text accordingly. Actually, I read the text a lot of times, but I did not see all the typos. Thanks!
4) Figure 1: It is almost identical to a figure to the figure of Feng et al. J. Cardiovasc. Dev. Dis. 2019. The authors must mention this.
The Reviewer is fully right! We clarified this source of inspiration and thank the Reviewer for the suggestion.
5) L 315: “The following influx of Ca2+ drives NLRP3 (nucleotide-binding domain, leucine-rich-containing family, pyrin domain-containing 3) inflammasome and IL-1b production, thereby favoring cardiac fibrosis [94].” In the mentioned reference, Ca2+i was not measured. Furthermore, it is known that it is rather the K+ efflux than Ca2+ influx that activates NLRP3.
We do thank the Reviewer for this recommendation. We have clarified this important issue mentioning the role of K+ efflux.
We therefore hope that the manuscript will now be regarded worth of being published on this thrilling special issue of Biomedicines.
Sincerely,
Francesco Moccia